3D bioprinting in bioremediation: a comprehensive review of principles, applications, and future directions

http://orcid.org/0000-0003-3435-0941 Finny Abraham Samuel 1 2 abrahamsamuel@gmx.com
1 Department of Chemistry and Biomolecular Science, Clarkson University , Potsdam, New York , United States
2 Waters Corporation , Milford, Massachusetts , United States
Azevedo Vasco
Electronic publication date: 2024 Feb 8
Publication date: 2024
Volume: 12
Electronic Location ID: e16897
Received 2023 Nov 20; Accepted 2024 Jan 16
Copyright: © 2024 Finny
Copyright year: 2024
Copyright holder: Finny
License: This is an open access article distributed under the terms of the Creative Commons Attribution License, which permits unrestricted use, distribution, reproduction and adaptation in any medium and for any purpose provided that it is properly attributed. For attribution, the original author(s), title, publication source (PeerJ) and either DOI or URL of the article must be cited.
License URL: https://creativecommons.org/licenses/by/4.0/

Keywords: 3D printing, Bioprinting, Bioanalysis, Analytical chemistry, Biosensing, Additive manufacturing, Chemistry, Bioremediation, Environment, Remediation

Funding: The author received no funding for this work.

==============================
Bioremediation is experiencing a paradigm shift by integrating three-dimensional (3D) bioprinting. This transformative approach augments the precision and versatility of engineering with the functional capabilities of material science to create environmental restoration strategies. This comprehensive review elucidates the foundational principles of 3D bioprinting technology for bioremediation, its current applications in bioremediation, and the prospective avenues for future research and technological evolution, emphasizing the intersection of additive manufacturing, functionalized biosystems, and environmental remediation; this review delineates how 3D bioprinting can tailor bioremediation apparatus to maximize pollutant degradation and removal. Innovations in biofabrication have yielded bio-based and biodegradable materials conducive to microbial proliferation and pollutant sequestration, thereby addressing contamination and adhering to sustainability precepts. The review presents an in-depth analysis of the application of 3D bioprinted constructs in enhancing bioremediation efforts, exemplifying the synergy between biological systems and engineered solutions. Concurrently, the review critically addresses the inherent challenges of incorporating 3D bioprinted materials into diverse ecological settings, including assessing their environmental impact, durability, and integration into large-scale bioremediation projects. Future perspectives discussed encompass the exploration of novel biocompatible materials, the automation of bioremediation, and the convergence of 3D bioprinting with cutting-edge fields such as nanotechnology and other emerging fields. This article posits 3D bioprinting as a cornerstone of next-generation bioremediation practices, offering scalable, customizable, and potentially greener solutions for reclaiming contaminated environments. Through this review, stakeholders in environmental science, engineering, and technology are provided with a critical appraisal of the current state of 3D bioprinting in bioremediation and its potential to drive forward the efficacy of environmental management practices.

Introduction

The United States Environmental Protection Agency (USEPA) reports that as of September 2023, 1,336 uncontrolled hazardous waste sites are registered on the National Priorities List (NPL) alone. The NPL is a crucial tool for identifying the contaminated sites requiring long-term remedial action through the Superfund program, and inclusion in the NPL is a critical step towards securing federal funding for the extensive cleanup operations required to remediate these hazardous waste sites (US Environmental Protection Agency (US EPA), 2023). Similarly, the Canadian government’s Federal Contaminated Sites Inventory has listed 4,503 active contaminated sites in Canada as of November 2023 (Environment and Climate Change Canada (ECCC), 2016; Treasury Board of Canada Secretariat, 2023). These numbers underscore the ongoing challenge of managing hazardous waste in North America. The European Union, for instance, is grappling with the daunting task of addressing pollution in approximately 2.8 million potentially affected land sites, as stated by the World Health Organization in July 2023 (World Health Organization (WHO), 2023). Meanwhile, the Global Alliance on Health and Pollution has identified over 5,000 toxic hotspots worldwide in low- and middle-income countries that require immediate remediation efforts (Global Alliance on Health and Pollution (GAHP), 2023). Therefore, addressing the significant environmental and public health risks posed by hazardous waste and contaminated sites remains an urgent and complex global issue that demands sustained commitment and resources.

Traditional remediation methods, such as excavation and incineration, can be expensive, generate hazardous waste, and have limited effectiveness. In contrast, bioremediation utilizes microorganisms or materials of biological origin, such as enzymes, biocomposites, biopolymers, or nanoparticles, to biochemically degrade contaminants into harmless substances, making it an environmentally friendly and cost-effective alternative. Bioremediation is a beacon of environmental sustainability, harnessing the power of biological processes and biomaterials to confront the escalating challenge of anthropogenic pollution. In the age where technological innovation is rapidly reshaping various industries, the field of environmental engineering is experiencing a renaissance with the advent of 3D printing technology, also known as additive manufacturing (Amorim et al., 2021; Gkantzou, Weinhart & Kara, 2023). This convergence can revolutionize bioremediation by offering novel solutions to complex environmental problems. 3D printing technology introduces unparalleled precision and customization to the fabrication of objects, operating under the principle of layer-by-layer construction from digital models. This technology is particularly promising for bioremediation, as it allows for the design and creation of intricate structures tailored to support microbial life or hold materials that are conducive to the removal of pollutants, facilitating the degradation of contaminants in diverse environmental matrices (Schubert, Van Langeveld & Donoso, 2014; Schaffner et al., 2017). The adaptability of 3D printing can be leveraged to enhance the efficiency of bioremediation strategies through the optimization of habitat architecture for microbial communities, thereby accelerating the biodegradation process. This confluence of biotechnology and additive manufacturing holds significant promise for developing innovative bioremediation strategies (Gross et al., 2014).

In recent years, there has been a significant surge of interest in using 3D printing and 3D bioprinting for bioremediation research. This is evident from the exponential increase in publications, with countries like China, USA, India, United Kingdom, Germany, and Spain leading the way (Fig. 1; Elsevier, 2023). These nations have invested heavily in advancing additive manufacturing technologies to support the development of cutting-edge bioremediation processes. Recent advancements in 3D printing have introduced materials and techniques specifically tailored for environmental applications. For instance, developing 3D-printed bioreactor media that can be customized to site-specific conditions, thereby maximizing microbial degradation activities, is a poignant illustration of the synergies between these technologies (Elliott et al., 2017). The high degree of customization enables the fabrication of structures with increased surface areas for microbial growth, optimizing the exposure of pollutants to degradative biofilms. By precisely controlling the spatial arrangement of cells and biomaterials, 3D bioprinting can create bioremediation devices with enhanced cell-cell interactions, improved nutrient and oxygen transport, and a more accurate representation of the physiological microenvironment, significantly enhancing their bioremediation performance (Chimene et al., 2016).

Figure 1 Number of publications between 2014–2023 using the combination of keywords “3D Printing” and “Bioremediation,” “3D Bioprinting” and “Bioremediation” found in Scopus.

The number of publications between 2014–2023 using the combination of keywords “3D Printing” and “Bioremediation,” “3D Bioprinting” and “Bioremediation” (A) and publications per country using the keywords keywords “3D Printing” and “Bioremediation,” (B) and keywords “Bioprinting” and “Bioremediation,” (C) found in Scopus (accessed on 10 Nov 2023; Elsevier, 2023).

This review critically examines the principles, applications, and future directions of 3D printing in bioremediation. By evaluating the current state of research, this article aims to provide insights into the potential environmental benefits and challenges associated with implementing 3D printing technologies in bioremediation.

Survey/search methodology

To ensure the inclusion of the most relevant and recent advancements, our search methodology encompassed a thorough literature review spanning the last two decades, focusing on publications from the last 5 years. Utilizing databases such as Scopus, Web of Science, PubMed, and Google Scholar, we employed keywords such as “3D Bioprinting,” “Bioremediation,” “3D Printing,” “Environmental Remediation,” and others as explained below to narrow down bioremediation research that utilized additive manufacturing processes. Priority was given to recent experimental and review articles that directly contribute to the understanding of 3D bioprinting applications in bioremediation, ensuring our review reflects the latest trends and technological developments in this rapidly evolving field. A strategic combination of keywords and Boolean operators was employed to provide a thorough and precise retrieval of pertinent literature. Primary keywords were initially used to identify relevant works and secondary key terms were used to ensure that potentially relevant results were not missed. Conventional search engines such as Google, Bing, and DuckDuckGo were also utilized to ensure recent non-indexed works were also captured.

Primary key terms included:

(“3D Printing” OR “Additive Manufacturing”) AND “Bioremediation”

“3D Bioprinting” AND “Bioremediation”

“Environmental Remediation” AND (“3D Printing” OR “Bioprinting”)

Examples of secondary key terms included but are not limited to:

“Biocarriers” AND “3D Printing”

“Enzyme Immobilization” AND (“3D Printing” OR “Bioprinting”)

(“3D Printing” OR “Bioprinting”) AND “Heavy Metal Remediation”

Inclusion and exclusion criteria

In order to maintain scholarly rigor, the following criteria were established:

Inclusion Criteria: Peer-reviewed articles published within the last two decades emphasizing the most recent 5 years to capture cutting-edge developments were included. Studies that explicitly discuss the utilization of 3D bioprinting within the scope of environmental bioremediation were prioritized. Additionally, we included papers contributing to the understanding of principles, applications, and prospective trajectories of 3D bioprinting in bioremediation.

Exclusion Criteria: Articles outside the realm of peer-reviewed literature were generally excluded, except where they provided unique and critical insights not available in peer-reviewed sources. Studies predating the 20-year window or those diverging from the core focus on bioremediation and 3D bioprinting were omitted.

Systematic selection process

Our literature search was executed in multiple phases to ensure depth and breadth. Initial searches using broad keywords yielded a diverse collection of articles, which were then scrutinized based on titles and abstracts for relevance. Subsequently, full-text assessments were conducted to ascertain the suitability of these studies against the defined inclusion criteria. To safeguard against selection bias and ensure a holistic perspective, the selection of articles was grounded in their scientific robustness and relevance to the subject matter, irrespective of their specific outcomes or the nature of their findings. Cross-referencing citations within these articles further augmented the breadth of our literature review. This exhaustive and methodically structured approach assured a nuanced and comprehensive review of the existing research landscape of 3D bioprinting in the context of bioremediation.

What is bioremediation?

Traditionally, bioremediation has encompassed using natural microorganisms or other life forms to accumulate and break down environmental pollutants to clean up contaminated areas. This includes methods such as natural attenuation, bioaugmentation, phytoremediation, and landfarming, among others. More recently, the term has been expanded to include techniques incorporating genetically modified organisms, biomaterials that mimic biological processes, and other customized approaches for environmental remediation. Examples of these techniques include pollutant degradation using genetically engineered organisms, bioventing, in situ bioreactors, and nanobioremediation. Throughout this article, we will use the term bioremediation to refer to any of the above-mentioned methods for environmental remediation.

What is 3D bioprinting?

3D bioprinting is an additive manufacturing technology that involves the precise layer-by-layer positioning of biological materials, biochemicals, and living cells to fabricate three-dimensional structures (Murphy & Atala, 2014). This approach provides spatial control of the placement of functional components, enabling the creation of complex and functional constructs. This field of bioprinting is a rapidly developing area that focuses on printing materials of biological origin, commonly referred to as bioinks (Fu et al., 2022), and while it has traditionally been applied in tissue engineering, the evolution from traditional to modern bioprinting techniques underscores significant technological advancements and a more comprehensive range of potential applications, including bioremediation, as explored in this article. This article will also show how biomaterials have been successfully incorporated into various conventional additive manufacturing technologies such as material extrusion, vat polymerization, powder bed fusion, material jetting, and binder jetting to create unique bioprinting tactics that could be used for bioremediation.

Considerations of 3D printing in bioremediation

Incorporating 3D printing technology into bioremediation practices offers several advantages that improve the efficacy and efficiency of conventional remediation methods. In this section, we will delve into the fundamental principles that govern the application of 3D printing technology in bioremediation efforts.

Design flexibility and customization

One of the critical advantages of utilizing 3D printing for bioremediation lies in the ability to create customized structures tailored to suit unique environmental conditions and contaminant profiles. This cutting-edge technology offers unparalleled control over the shape, size, and surface area of the printed objects, thereby enabling the design of intricate features that can enhance microbial colonization and the efficient degradation of pollutants (Duty et al., 2017). Additive manufacturing is revolutionizing the manufacturing industry by allowing the creation of highly customized designs and enabling rapid prototyping (Strack, 2019). In bioremediation, this technology could facilitate scale-up and mass production while simultaneously reducing costs, lead times, and waste. Additionally, additive manufacturing uses less energy than traditional manufacturing processes, making it a more sustainable solution for various environmental remediation techniques.

Material selection and sustainability

The use of 3D printing technologies has made it possible to incorporate a wide range of materials, such as biopolymers and recycled plastics, into the printing process. These materials are carefully chosen based on their ability to be printed, compatibility with biological systems, and ability to decompose naturally. By selecting sustainable materials, the 3D-printed structures themselves do not contribute to pollution but rather contribute to the overall remediation process (Zhang et al., 2023). An example is the inherent challenge in integrating functional bacteria with 3D bioprinting, which lies in achieving a delicate equilibrium between the manufacturability of the material, minimizing damage during the bioprinting procedure, and preserving bacterial activity and function (Zhao et al., 2023). Optimizing bioink selection, considering 3D printability, microbial and chemical compatibility, and contaminant degradation, is crucial for bioremediation.

Compatibility of the materials with microbial systems is, therefore, an essential criterion for material selection, which facilitates successful integration with microbial systems.

The design of 3D-printed structures for microbial-based bioremediation is closely aligned with the biological requirements of the microorganisms involved. This involves considering factors such as nutrient flow, aeration, and maintaining optimal environmental conditions for microbial growth and enzymatic activity (Cao et al., 2022). Using 3D-printed biocarriers has been shown to improve the nitrification efficiency of designed systems by facilitating the growth of sluggish bacterial species, highlighting the importance of ensuring microbial compatibility when considering their effectiveness for remediation purposes (Noor et al., 2023). It has been established that successful 3D printing of living materials with high performance relies on the development of new ink materials and 3D geometries that promote long-term cell functionality (Qian et al., 2019).

Increased surface area

The effectiveness of bioremediation processes is often influenced by the surface area available for microbial activity. 3D printing makes it possible to create porous structures with a favorable surface-to-volume ratio, allowing for significant space for chemical or microbial reactions. This innovative technology also boosts the accessibility of pollutants for degradation on functional surfaces (Aguirre-Cortés et al., 2023). Bioprinting can provide high surface area per unit volume, lightweight structure, high porosity, and roughness that are essential for the growth of biofilms, a widely used biocarrier for bioremediation purposes (Sfetsas, Patsatzis & Chioti, 2021).

Scalability and reproducibility

3D printing technology enables the production of bioremediation tools that can be scaled from small laboratory prototypes to larger structures suitable for field applications. Moreover, the digital nature of 3D printing ensures reproducibility, allowing for the consistent manufacture of bioremediation devices across different locations (Thompson et al., 2016). Recent research has identified key factors that can be improved to enhance the reproducibility and reliability of bioprinting, which holds great potential for future applications (Grijalva Garces et al., 2024).

Life-cycle and ecological impact assessment

A principle that is becoming increasingly significant in the application of 3D printing to bioremediation is the assessment of the life-cycle impact of the printed structures. Evaluating the ecological footprint of these materials from production to degradation is crucial for ensuring that the bioremediation strategy is truly sustainable (Roy et al., 2009). Life cycle assessment is a comprehensive approach that considers various factors, such as the type and amount of raw materials used, energy consumption throughout the technology/activity’s life cycle, and the amount of waste released to the environment. It aims to evaluate and quantify the environmental impact associated with a particular technology/activity in a detailed and rigorous manner. Additive manufacturing is a cost-effective solution for producing intricate and lightweight geometries, particularly in small batch quantities and situations where it can reduce lead times, which is highly relevant for bioremediation, but its overall economic potential could still be limited by factors such as expensive printers, lower production capacity, and slower build rates, and the societal impact of additive manufacturing on various stakeholders, including workers, local communities, society, consumers, and value chain actors, has yet to be fully assessed as this research is still in its early stages (Kokare, Oliveira & Godina, 2023).

Multi-material and function integration

Modern 3D printing technologies can incorporate multiple materials into a single print, creating complex devices with integrated functions such as embedded sensors for monitoring remediation progress or channels for optimized distribution of nutrients and microorganisms (Nazir et al., 2023). A system for modeling microbes in 3D geometries using projection SLA to bioprint microbes within hydrogel matrices was able to show promise for engineering biofilms with dual functionality: metal sequestration and the uranium sensing capability using Caulobacter crescentus strains (Dubbin et al., 2021). A previous study used a gelatin/alginate (5% and 2% w/v, respectively) biomaterial ink containing B. subtilis 2569 that was genetically tailored to fabricate engineered multifunctional biofilms for fluorescence detection, conjugation chemistry, single-substrate bioremediation, and multi reaction bioremediation cascades incorporating nanoparticles (Huang et al., 2019).

Therefore, utilizing the capabilities of 3D printing technology has the potential to transform the field of bioremediation by offering improved, innovative, and practical solutions for addressing environmental contamination. By adopting combinative bioremediation approaches such as this, we can fully leverage the ability of various bioremediation strategies and effectively combat environmental challenges with greater efficiency.

Additive manufacturing techniques in bioremediation

3D printing encompasses a range of technologies that create objects by adding material layer-by-layer based on digital models. Each of these technologies offers distinctive benefits that can be leveraged in bioremediation to develop structures with characteristics tailored to environmental cleanup needs. Even though the adoption of this technology in bioremediation is in its early stages, the area is seeing massive growth. Below is a description of several key 3D printing methods, how they can be relevant to bioremediation, and examples where they are used for remediation purposes. Figure 2 depicts the specific 3D bioprinting methods utilized in bioremediation, highlighted in green.

Figure 2 3D bioprinting methods utilized in bioremediation.

Stereolithography (SLA)

SLA is one of the oldest and most precise 3D printing techniques, which uses an ultraviolet (UV) laser to cure and solidify photopolymer resin layer by layer. SLA leverages photopolymerization, wherein a vat of liquid photopolymer resin cures upon exposure to a targeted UV light source. The resins typically blend reactive monomers and oligomers, photoinitiators, and various functional additives. During printing, photoinitiators absorb UV light to generate reactive species (free radicals or cations) that propagate a chain reaction, leading to crosslinking and forming a solid polymer matrix. The high resolution of SLA can produce parts with smooth surface finishes and intricate details. In bioremediation, the precision of SLA is particularly beneficial for creating microfluidic devices used in lab-on-a-chip applications that simulate environmental conditions for research and development of remediation strategies (Huang et al., 2015). In the context of bioremediation, SLA can be used as a tool for creating the necessary structures and devices to support the growth and activity of biological agents that break down pollutants (Kadiak & Kadilak, 2017; Liu et al., 2023b). A recent study reported on the development of SiO2/TiO2/polymer scaffolds using SLA technology. These scaffolds incorporated sugarcane leaf-derived SiO2 as the adsorbent, multi-phase TiO2 synthesized through a solution combustion technique as the photocatalyst, and a photocurable resin as the structural material. The scaffolds demonstrated an average total removal efficiency of 81.9% for methylene blue and 60% for rhodamine B dyes, which shows potential for use in wastewater treatment applications (Bansiddhi et al., 2023). However, the practicality of using SLA-printed items in bioremediation depends on developing and using appropriate materials that align with environmental safety and sustainability goals.

Selective laser sintering (SLS)

SLS uses a laser to sinter powdered material, typically nylon or polyamide, to form solid structures. This method can produce durable and complex geometries without supporting structures. SLS has also been used to manufacture structurally complex miniaturized photobioreactor parts using polyamide (Krujatz et al., 2016). SLS-printed parts can fabricate sturdy components for bioremediation processes that may require good mechanical properties and chemical resistance from harsh environmental conditions (Hopkinson, Hague & Dickens, 2006). In recent times, there has been an increased interest in the development of polymeric nanocomposites for water treatment applications using SLS to create durable, efficient, and cost-effective polymer nanocomposites that are monodisperse, highly reactive and have minimal surface or structural defects (Adeola & Nomngongo, 2022).

Fused deposition modeling (FDM)

FDM, also known as fused filament fabrication (FFF), is a widely used 3D printing method that extrudes thermoplastic polymers through a heated nozzle to form layers. FDM is highly versatile and allows for the printing of large parts at a lower cost. For bioremediation, FDM can be utilized to create custom housings for biofilters or frameworks for biofilm reactors that are scalable and cost-effective (Rocha et al., 2017). Studies using natural biopolymers and biopolymer-based materials, including chitosan, polylactic acid (PLA), alginate, and cellulose acetate (CA), for potential applications within the water treatment industry with emphasis on oil separation and metal removal, are being done using FDM (Fijoł, Aguilar-Sánchez & Mathew, 2022).

Digital light processing (DLP)

Similar to SLA, DLP 3D printing also uses a light source to cure photopolymers, but it does so by projecting an entire layer’s image at once, which can result in faster print times. DLP is particularly suited for manufacturing small to medium-sized intricate structures that require high precision, such as scaffolds for microbial attachment in bioremediation systems (Melchels, Feijen & Grijpma, 2010). Recently, researchers have been working on developing a platform for extrusion 3D bioprinting of hydrogel-based bio-inks loaded with diatoms, where a digital light processing (DLP) bioprinting platform was used to shape photolabile polymers containing dinoflagellates or diatoms that were responsive to contaminants (salt, antimicrobial agents and herbicide), even though this was developed for biosensing, platforms such as this could be easily adapted to created biohybrid materials that could be used for bioremediation (Boons et al., 2023).

Material jetting and binder jetting

Material Jetting involves jetting droplets of photopolymer, which are then cured by UV light. It is known for its ability to produce parts with high accuracy and smooth surfaces and its capacity to print with multiple materials simultaneously. This could be advantageous for creating multi-material bioremediation devices with structural and functional elements integrated into a single print (Derby, 2010).

Binder jetting involves selectively depositing a liquid binding agent onto a powder bed, bonding these areas together to form a part. Since it can use various materials, including metals, sands, and ceramics, this method can produce components for bioremediation that require specific material properties, such as catalyst supports for the chemical degradation of pollutants (Gibson, Rosen & Stucker, 2015). These techniques are used in tandem with other additive manufacturing technologies for bioremediation applications.

Multi-material 3D bioprinting

Advanced 3D bioprinting technologies can handle multiple materials within a single printable bioink formulation. This allows for the fabrication of complex devices with varying material properties, including combining biodegradable materials with functional additives that enhance microbial growth or pollutant adsorption in bioremediation processes (Sun et al., 2013). Bioprinting, a specialized form of multi-material 3D printing that is extrusion-based, involves the precise layering of bioinks composed of cells, growth factors, and functional biomaterials to construct biofunctional structures. For experts in bioremediation, bioprinting opens a frontier for fabricating bio constructs tailored to degrade environmental pollutants. These living or biochemically functional architectures can be engineered to optimize the viability, functionality, and performance of the encapsulated particles, which may be cells, nanoparticles, enzymes, or other functional materials, thereby enhancing the efficiency and specificity of biodegradation pathways. By manipulating the composition and spatial distribution of different cell types within a bioink, bioprinted constructs can be customized to target specific contaminants. Additionally, integrating sensing components within bioprinted matrices can lead to the development of intelligent bioremediation systems capable of real-time monitoring and response. Advancements in bioink development, focusing on immobilizing microbes, enzymes, nanoparticles, metal-organic frameworks, or particles with catabolic prowess, are pivotal for extending bioprinting applications towards eco-restoration and pollution abatement.

For bioremediation, these additive manufacturing techniques can be strategically selected based on the requirements of the remediation task, such as biodegradability, biocompatibility, chemical resistance, mechanical properties, and the complexity of the structures required for optimizing the degradation of contaminants. Table 1 provides a quick snapshot of how each of the above techniques could potentially play a role in bioremediation and their relative advantages and disadvantages. As the demand for innovative and sustainable bioremediation solutions continues to grow, the potential of various additive manufacturing technologies to revolutionize the way we tackle bioremediation is becoming increasingly evident and poised to become a key player in developing next-generation bioremediation techniques.

Table 1 Comparative view of different 3D printing techniques in the context of bioremediation.

3D printing technique	Material compatibility	Structural complexity	Durability in tough environmental conditions	Economic feasibility	Suitability for bioremediation	References	
Stereolithography (SLA)	Photopolymers	High	Moderate	Moderate-high	Suitable for intricate bio-scaffolds and microfluidic devices for controlled bioremediation environments	Huang et al. (2015), Kadiak & Kadilak (2017), Liu et al. (2023b), Bansiddhi et al. (2023)	
Selective laser sintering (SLS)	Nylon, polyamide	Moderate-high	High	Moderate-high	Ideal for producing robust components for harsh environmental conditions	Krujatz et al. (2016), Hopkinson, Hague & Dickens (2006), Adeola & Nomngongo (2022)	
Fused deposition modeling (FDM)	Thermoplastics (e.g., PLA, ABS)	Moderate	Moderate	Low-moderate	Cost-effective for large-scale bioreactors and support structures	Rocha et al. (2017), Fijoł, Aguilar-Sánchez & Mathew (2022)	
Digital light processing (DLP)	Photopolymers	High	Moderate	Moderate-high	Suitable for precision structures like biofilm scaffolds and micro-bioreactors	Melchels, Feijen & Grijpma (2010), Boons et al. (2023)	
Material jetting	Photopolymers, waxes	High	Moderate	High	Applicable for multi-material structures with gradients for selective bioremediation	Derby (2010)	
Binder jetting	Metals, sands, ceramics	Moderate	High	Moderate	Useful for creating catalyst supports and filtration systems in water and soil remediation	Gibson, Rosen & Stucker (2015)	

Materials for 3D printing in bioremediation

The selection of materials in 3D printing for bioremediation is critical, as the materials must not only be suitable for the printing process but also conducive to bioremediation activities. For example, when used in bioremediation, certain materials must support microbial life for the degradation of pollutants and be mechanically stable while also being environmentally sustainable. Exploring new materials and techniques to achieve efficient and cost-effective bioremediation processes is also crucial. Some common materials for 3D bioprinting that could play a role in bioremediation are discussed below with examples.

Biodegradable polymers

Biodegradable polymers are favored in bioremediation applications for their ability to break down naturally over time, minimizing environmental impact. Polylactic acid (PLA) is one such polymer, popular in 3D printing for its ease of use and compostable properties. PLA can be used to create frameworks for microbial films in water treatment or soil remediation, gradually degrading into harmless lactic acid (Farah, Anderson & Langer, 2016). Another research work using PLA has led to the creation of a bioremediation system based on using a native isolate of Chlorella vulgaris immobilized onto an alginate matrix inside a PLA device, where the researchers were able to successfully demonstrate the reduction of all inorganic nitrogen forms and total phosphorus by 90% after 5 days, and a 85% decrease in aerobic mesophilic bacteria (Marconi et al., 2020). Polyhydroxyalkanoates (PHAs) are another class of biopolymers produced by bacterial fermentation of sugars or lipids and are completely biodegradable, making them ideal for temporary structures in ecosystem restoration projects (Kourmentza et al., 2017). Ongoing research in this field will likely lead to even more innovative uses for these materials in the future.

Composites

Composites that blend biodegradable polymers with natural fibers or fillers can enhance the mechanical properties and biodegradability of printed objects. For instance, a composite of PLA and natural fibers like cellulose can be designed to provide structural support in bioremediation systems while maintaining biodegradability (Benini, de Bomfim & Voorwald, 2023). In another recent work, researchers combined microgel-based granular inks that were 3D printable to fabricate bacteria-induced biomineral composites that were biomimetic comprising 93wt% calcium carbonate and the ability to withstand pressures up to 3.5 Mpa (Fig. 3, reused with permission from Hirsch et al., 2023) for potential use as artificial corals to help in the regeneration of marine reefs and ocean remediation applications (Hirsch et al., 2023). Nanocellulose is another popular choice for creating 3D printable functional composites (Finny, Popoola & Andreescu, 2021). 3D printable oil/water separators that could act as sponges to remove oil and other microorganisms from polluted sites have been developed using nanocellulose composites (Firmanda et al., 2023). 3D printable composites using polycaprolactone (PCL) and sodium alginate were found to have heavy metal adsorption properties, and the authors were able to demonstrate that sodium alginate retained its heavy metal adsorption properties within the PCL filament and was able to remove 91.5% of copper ions from a 0.17% w/w copper sulfate solution in 30 days thus making thermoplastic composite filaments such as these an exciting option for complex contaminated sites needing tailored solutions (Liakos et al., 2020). From the examples above, it is clear that the development of such functional composites opens up exciting bioremediation possibilities for tailored solutions in complex contaminated sites.

Figure 3 3D printed biomineral composites.

(A) Formulation of the bioink. (B) 3D printing. (C) Mineralization. (D) Schematic representation of the microbially-induced calcium carbonate precipitation process mediated by S. pasteurii. (E) 3D printed biomineral composite after four days of microbially-induced calcium carbonate precipitation process; the displayed scale bar is 10 mm. Image source credit: Hirsch et al. (2023), CC BY 4.0 DEED, https://creativecommons.org/licenses/by/4.0/.

Functionalized materials

Functionalized materials that contain adsorbents like activated carbon can be used to fabricate filters and membranes. These specialized materials are engineered to capture specific pollutants while allowing the proliferation of microorganisms that can degrade these pollutants (Fan et al., 2022). Researchers have constructed a 3D printing platform that uses rudimentary alginate chemistry for printing a bacteria-alginate bioink mixture onto calcium-containing agar surfaces, which resulted in the formation of bacteria-encapsulating hydrogels with varying geometries with the potential to be used as biofilms for environmental detoxification purposes such as bioremediation, heavy metal removal, removal of assimilable organic carbon, and wastewater treatment (Balasubramanian, Aubin-Tam & Meyer, 2019). A functional material encapsulating Pseudomonas putida, a bacteria, in a biocompatible and functionalized 3D printable ink consisting of sodium hyaluronate and glycidyl methacrylate to print a “living material” capable of degrading phenol, a common pollutant, was demonstrated to show total phenol degradation after 40 h into harmless biomass (Schaffner et al., 2017).

The ability of hydrogels to form hydrophilic aqueous microenvironments maintaining the reactivity of various catalysts along with their advantageous properties such as biocompatibility, swelling ability, and resistance to dissolution, make hydrogels ideal candidates for bioimmobilization and functionalization, as they provide improved stability of the immobilized components preventing leakages and the diffusion of substrate molecules and their reaction products. Sodium alginate and bentonite clay were used to create 3D printable nanocomposite hydrogels for the adsorption of the pesticide paraquat, and the removal tests indicated that the adsorption process was due to spontaneous adsorption mechanisms involving physisorption, showing a maximum adsorption capacity at equilibrium of 2.29 mg/g with an ability to be reused for at least six cycles (Baigorria et al., 2023). Overall, these examples demonstrate the promising potential of functionalized materials and hydrogels in environmental remediation. By using specialized engineered materials and 3D printing technology, researchers can create innovative solutions for pollutant removal and wastewater treatment. These advancements in materials science and biotechnology offer hope for a cleaner and more sustainable future.

Even though various materials could be used for environmental remediation, materials for 3D printing in bioremediation must be carefully chosen to ensure they do not introduce new contaminants, support the life cycle of the encapsulated or immobilized biological/chemical component, and have a negligible environmental footprint after their useful life.

Design and modeling

The conceptualization and execution of 3D structures tailored for bioremediation necessitate an interdisciplinary collaboration of environmental engineering, chemistry, biology, and materials science expertise. This intricate design and modeling process must encapsulate the multifaceted interactions between biological consortia and their physicochemical surroundings, ensuring that the created habitats not only foster microbial growth but also provide an active environment for the encapsulated active degradants to biodegrade the pollutants optimally. In a study aimed at removing drugs from water, researchers fabricated a device using SLA where they immobilized laccase sourced from Trametes Versicolor within a poly(ethylene glycol) diacrylate hydrogel and found that when the device was configured in the shape of a torus, it removed 95% of diclofenac and ethinylestradiol from aqueous solution within 24 and 2 h, respectively, and was much more efficient than free enzyme (Xu et al., 2022a). This highlights the significance of creating and tuning optimal material geometries that favor pollutant removal when fabricating adsorbents.

Computational design

Computational tools are essential in the design process, enabling the simulation of different scenarios and the optimization of structures for maximum efficiency. Software such as computer-aided design (CAD) programs allows the creation of detailed 3D models that can be tested virtually under different conditions. Computational fluid dynamics (CFD) can simulate the flow of water or air through the structures, helping optimize nutrient distribution and waste removal, essential factors for microbial growth (Versteeg & Malalasekera, 2007). Bioprinting is also fundamentally interdisciplinary, and therefore, it provides an opportunity for scientists and engineers to collaborate to apply engineering design and standardization parameters to the printing and analysis processes (Correia Carreira, Begum & Perriman, 2020). Therefore, the incorporation of this technique into bioremediation methodology provides an avenue for pioneering interdisciplinary investigations.

Design considerations

When designing 3D structures for bioremediation, considerations include maximizing surface area for microbial colonization, enzyme/particle reactivity, and creating pore sizes that allow optimal flow rates and structural integrity to withstand environmental stresses. The design must also account for the ease of scaling up from laboratory to field sizes and the adaptability to different pollutants and ecological conditions (Pant et al., 2010). Integrating advanced design and modeling techniques ensures that 3D printed structures for bioremediation are optimized for environmental applications, promoting effective pollutant degradation and efficiency.

Bio-inspired design

Bio-inspired design, which emulates natural structures such as honeycombs or plant roots, can be particularly effective in bioremediation. These structures can be modeled to create complex geometries that mimic biological systems, offering high surface areas and efficient nutrient distribution pathways for microorganisms (Wang, Chen & Chen, 2020). Research on bioinspired nanosurfaces with tailored multifunctionality, such as hydrophobicity, has attracted significant attention for scientific exploration and practical applications inspired by natural phenomena. As a result, 3D printing has emerged as an up-and-coming method for producing biomimetic materials with diverse applications due to its numerous advantages, including customizability, affordability, and accessibility (Wang et al., 2023a).

Applications of 3D printing in bioremediation

This section delves into the ways in which 3D printing can be used in bioremediation, exploring topics such as microbial support structures, enzyme immobilization, heavy metal adsorption and filtration, case studies, and potential obstacles.

Microbial support structures

The success of bioremediation often hinges on the health and stability of microbial colonies. 3D printing has revolutionized the development of microbial support structures by enabling the creation of complex geometries tailored for microbial growth. These structures are designed to provide a high surface area-to-volume ratio, which is crucial for the colonization and bioactivity of microbes. Research has demonstrated that the porosity and the interconnectivity of the pores can be finely tuned to control the distribution of nutrients and the removal of metabolites, thus optimizing the bioremediation process (Bhattacharjee et al., 2016). Studies focusing on water treatment have utilized 3D-printed lattice structures that facilitate the growth of biofilms, which are integral in the degradation of organic pollutants (Dzionek, Wojcieszyńska & Guzik, 2016). Researchers have formulated a dual-network bioink for 3D printing of “living materials” with enhanced biocatalysis properties, where the printable bioinks provide a biocompatible environment along with desirable mechanical performance; integrating microbes into these bioinks enabled the direct printing of catalytically living materials with high cell viability and optimal metabolic activity, for potential use in the bioremediation of chemicals; this study showed more than 90% degradation of methyl orange and acrylamide in 48 h using a bacteria-microalgae within the bioink matrix (He et al., 2022). A novel dual-crosslinking poly(ethylene glycol) diacrylate-alginate-poly (vinyl alcohol)-nanoclay (PAPN) bio-ink containing one heterotrophic bacterium (Oceanimonas sp. XH2) was reported, where the authors used extrusion-based 3D printing to create a functional biomaterial with the capabilities of ammonia removal; the authors showed that the 3D printed PAPN functional material could remove 96.2 ± 1.3% ammonia within 12 h, and they also observed that the removal rate of ammonia increase with repeated use due to the rise in bacteria within the bio-scaffolds over time (Li et al., 2022). Similarly, various bacterial and microbial species are now being mixed with polymers creating functional complex bioinks, and these systems show enormous potential in applications such as bioremediation, and sometimes they can even respond to pollutants serving as sensors that can detect toxic chemicals and also potentially as oil spill filters as discussed earlier, making these 3D printed “minibiofactories” outstanding candidates for biotechnology-based bioremediation (Kyle, 2018). Researchers have recently introduced a new micromodel technology that has been designed to investigate bacterial biofilm formation in porous media. This technology is particularly useful for understanding biofilm dynamics in various applications, including wastewater treatment and soil bioremediation. The heart of this technology is a 3D-printed micromodel that enables the growth of biofilm within a perfusable porous structure. By utilizing high-precision additive manufacturing techniques, particularly stereolithography, the authors have developed a system that allows for precise control over the microenvironment, including flow channels and substrate architecture. One of the key advantages of this technology is the ability to monitor crucial parameters such as oxygen consumption, pressure changes, and biofilm detachment, which are essential for comprehending and optimizing biofilm behavior. The authors have demonstrated how this technology can be used to study Pseudomonas aeruginosa biofilm development for several days within a network of flow channels (Papadopoulos et al., 2023). Studies like these demonstrate the benefits of using additive manufacturing techniques to create consistent 3D porous microarchitectures, and these approaches act as ideal platforms for examining the dynamics of biofilm development in 3D porous media and quickly refining processes that promote bioremediation.

Enzyme immobilization

The field of enzyme immobilization has greatly benefited from the advent of 3D printing. The technique allows for the precise placement of enzymes on various substrates, which can be used to catalyze the breakdown of pollutants. This spatial control not only improves the stability and reusability of enzymes but also enhances the efficiency of the bioremediation process. Researchers have leveraged 3D printing to develop bioreactors where enzymes are immobilized on printed scaffolds, resulting in increased degradation rates of pollutants like phenol and other aromatic compounds (Shao et al., 2022; Bellou et al., 2022). 3D printing has also been used to create an enzyme-immobilized platform for biocatalysis by formulating a printable hydrogel ink comprising of dimethacrylate-functionalized Pluronic F127 (a non-ionic copolymer surfactant) and sodium alginate with the enzyme laccase for possible uses in environmental remediation. A piece of work using 3D bioprinting utilized a bioink made of sodium alginate, acrylamide, and hydroxyapatite with immobilized laccase for biodegradation of p-chlorophenol where the immobilized laccase exhibited excellent storage stability and reusability and retained over 80% of its initial enzyme activity after three days of storage, and was able to be reused for treating seven batches of phenolic compounds (Liu et al., 2020). Another recent work using laccase reported a biocatalytic system using immobilized laccase to 3D printed open-structure biopolymer scaffolds that were shown to remove 35-40% of estrogen group hormones such as 17β-estradiol and 17α-ethynylestradiol from municipal wastewater containing 56 ng/L of 17α-ethynylestradiol and 187 ng/L of 17β-estradiol (Rybarczyk et al., 2023). Research has also demonstrated that these estrogen group hormones could bind onto 3D-printed (SLS) filters made from commonly used polymers, such as polyamide-12 (PA), thermoplastic polyurethane (TPU), polypropylene (PP), and polystyrene (PS), and these filters showed enhanced surface morphology (Fig. 4, reused with permission from (Frimodig & Haukka, 2023)) and removal capacities of 35, 32 and 37 μg g−1 for estrone, 17β-estradiol, and 17α-ethinylestradiol, respectively (Frimodig & Haukka, 2023). These developments underscore the potential of 3D printing in creating more effective bioremediation tools and systems.

Figure 4 Surface of each 3D printed filter imaged using SEM.

Scanning electron microscope images of surfaces of each 3D printed filter using (A) polyamide-12, (B) polystyrene, (C) thermoplastic polyurethane, (D) polypropylene, (E) A simplified cross-section of the 3D-printed material illustrating solvent flow-through; notice the abundance of available surface area. Image source credit: Frimodig & Haukka (2023), CC BY 3.0 DEED, https://creativecommons.org/licenses/by/3.0/.

Heavy metal adsorption and filtration

Heavy metal contamination is a critical environmental issue, and 3D printing has emerged as a promising approach to developing novel adsorption and filtration systems. For example, a study that looked at sediment samples collected from three locations in Port Everglades, Florida, USA, indicated elevated ecological risk because of moderate-to-significantly high heavy metal contamination [As (0.607–223 ppm), Cd (n/d–0.916 ppm), Cr (0.155–56.8 ppm), Co (0.0238–7.40 ppm), Cu (0.004–215 ppm), Pb (0.0169–73.8 ppm), Mn (1.61–204 ppm), Hg (n/d–0.736 ppm), Mn (1.61–204 ppm), Ni (0.232–29.3 ppm), Se (n/d–4.79 ppm), Sn (n/d–140 ppm), V (0.160–176 ppm), and Zn (0.112–603 ppm); n/d = not-detected] (Giarikos et al., 2023). 3D-printed structures can be embedded with materials like biochar, activated carbon, or metal-organic frameworks, which have a high affinity for heavy metals and can serve as potential remediation solutions for such issues. The design flexibility of 3D printing allows for the optimization of these structures, maximizing contact time and enhancing the removal efficiency of heavy metals such as lead, cadmium, and arsenic from contaminated water and soil (Ignatyev, Thielemans & Vander Beke, 2014; Fee, Nawada & Dimartino, 2014). Researchers have also reported a polylactic acid-hydroxyapatite biocomposite prepared through a solvent-assisted blending and thermally induced phase separation technique, which was processed into highly permeable 3D biofilters using FDM showing maximum adsorption capacities of 112.1 and 360.5 mg/g for the metal salts of lead and cadmium respectively (Fijoł et al., 2021).

A new work using a chitosan-hydroxyapatite coupled with PLA to create monolithic filters utilizing 3D printing demonstrated robust Cu2+ removal performance with a maximum adsorption capacity of 119 mg/g, exhibiting the ability to remove more than 80% Cu2+ from their sample in less than 35 min (Wang et al., 2023b). The graphical representation of how the authors fabricate the filter can be seen in Fig. 5 (Reused with permission from (Wang et al., 2023b)). A previous work where a reusable monolithic 3D porous adsorbing filter was 3D printed using chitosan for heavy metal removal showed an adsorption capacity of 13.7 mg/g with adsorption kinetics of 2.2 mg/m per minute for Cu2+ removal and this is further proof for the role of 3D bioprinting in the field of bioremediation (Zhang et al., 2019).

Figure 5 Graphical representation of the PLA-Chitosan(CS)-Hydroxyapatite(HAP) filter fabrication.

Image source credit: Wang et al. (2023b), CC BY 4.0, https://creativecommons.org/licenses/by/4.0.

Multifunctional, robust, reusable, and high-flux filters are needed for sustainable water treatment and bioremediation, and to accomplish this, biobased and biodegradable water purification filters were developed and processed through 3D printing, more specifically using FDM; here, the authors used polylactic acid (PLA) based composites reinforced with homogenously dispersed (2,2,6,6-Tetramethylpiperidin-1-yl)oxyl -oxidized cellulose nanofibers (TCNF) and chitin nanofibers (ChNF), and they have an adsorption capacity towards copper ions as high as 234 (TCNF) and 208 mg/g (ChNF) and maximum separation efficiency of 54% (TCNF) and 35% (ChNF) towards microplastics in laundry effluent water (Fijoł et al., 2023a). 3D printing has also been combined with surface segregation and vapor-induced phase separation process to create structured adsorbents using composite inks consisting of polysulfone, polystyrene-block-poly(acrylic acid) and carbon nanotubes coupled with poly(ethyleneimine) (PEI) and terpyridine-COOH to get sorbents with copper ion removal capabilities of up to 31.3 mmol/m2; however, they observed degradation in copper removal in the presence of other ions (Xu et al., 2022b).

Biopolymer-based 3D printable hydrogels have also been explored for heavy metal removal from water, where a bioink consisting of shear-thinning hydrogels was fabricated by mixing chitosan with diacrylated Pluronic F-127, which showed 95% metal removal within 30 min in some cases (Appuhamillage et al., 2019). A one-step 3D printing method (Fig. 6, reused with permission from (Finny et al., 2022)) using 3D printable hydrogel-based adsorbents using alginate, gelatin, and polyethyleneimine-based bioink has also been reported to show excellent heavy metal ion removal adsorption capacities of 90.38%, 59.87%, 46.27%, 38.66%, and 6.45% for Cu2+, Ni2+, Cd2+, Co2+, and Pb2+ ions respectively from the tested samples (Finny et al., 2022). A 3D printable nanocomposite hydrogel was fabricated through electron beam crosslinking of alginate/nanoclay to remove inorganic micropollutants from wastewater for heavy metal removal applications where the authors note a maximum removal capacity of 532 mg/g for Pb(II) ions (Shahbazi et al., 2020).

Figure 6 One-step 3D-printing of heavy metal removal hydrogel tablets.

Illustration of the one-step 3D-printing fabrication (A) and removal (B) process of the hydrogel tablets, showing the interaction between PEI and Cu2+ ions, as an example. The hydrogel turns blue in the presence of Cu2+ due to the chelation process leading to the formation of cuprammonium complexes within the printed hydrogel. (Reused with permission from Finny et al., 2022).

Hydrogel filters containing algae cells have been 3D printed and experimentally shown to remove copper from test solutions by about 83% in 1 h (Thakare et al., 2021). A 3D-printed monolith fabricated using DLP using polyethylene glycol diacrylate, a plant-based resin, and chitosan exhibited removal efficiencies of 20.8 % to 90.4 % for methyl orange dye with an equilibrium uptake capacity ranging from 1 to 12.7 (mg/g) after 2 h (Husna et al., 2022). In a recent work, cellulose and metal-organic frameworks were combined to create a 3D printed composite material that exhibited CO2 and heavy metal ions adsorption capacities of 0.63 mmol/g (27.7 mg/g) and 8 to 328 mg/g, respectively while also displaying complete (>99%) removal of organic dyes in 10 min with high selectivity toward anionic dyes like methylene blue (Nasser Abdelhamid, Sultan & Mathew, 2023). 3D printed biobased filters anchored with a green metal-organic framework have shown to have maximum adsorption efficiencies of 42.3% for Pb (II), 72.8% for Mn (II), 21.1% for As (III), 47.1% for Cd (II) and 41% for Zn (II) after 24 h, making them potential candidates for effluent treatment (Fijoł et al., 2023b). Polydopamine (PDA) and bovine serum albumin (BSA) were added to a graphene-based ink to 3D print graphene-biopolymer aerogels for water contaminant removal as a proof of concept and preliminary results showed that the aerogel removed 100% organic solvents over 10 cycles of regeneration and reuse (Masud, Zhou & Aich, 2021). Researchers have recently created recyclable 3D printed hydrogel composites that incorporate biochar sourced from rice husk for removing organic contaminants from tap water and have experimentally demonstrated that the hydrogel containing 10% w/w biochar (Alginate/Biochar) demonstrated significant adsorption capacities of 111.4 mg/g for ibuprofen (IBU) and 214.6 mg/g for methylene blue (MB) which represents an increase in adsorption capacities of 48% (IBU) and 58% (MB) compared to conventional hydrogels without biochar. This innovative development highlights the potential of novel composites and underscores the importance of continuing to explore new avenues for improving water quality (Silva et al., 2023).

Case studies

Real-world case studies illustrate the practical applications of 3D printing in bioremediation. One such example is the deployment of 3D-printed biofilters for the treatment of industrial wastewater, where the specificity of the printed matrix improved the reduction of nitrogen and phosphorus levels (Mohd Yusoff et al., 2023). Another case involved using 3D-printed sponges for oil spill management, where the porous structures enhanced the absorption of hydrocarbons, facilitating the subsequent biodegradation by marine microbes (Walker & Humphries, 2019). These case studies, paired with the multiple works discussed previously, showcase the potential of 3D printing technology in environmental remediation. The ability to customize the matrix of the printed materials offers a high degree of control in designing effective and efficient bioremediation systems; the success of these case studies provides a promising outlook for the future of 3D printing in bioremediation and highlights the importance of interdisciplinary collaborations between engineering and environmental science. Analyzing patents could also be indicative of the commercialization potential for these technologies and could provide valuable insights. The University of Rochester has patented a low-cost and efficient 3D printing method for creating genetically modified Escherichia coli biofilms, which can be used for environmental detoxification and bioremediation (Meyer, 2020). Princeton University has filed a patent for a method of manufacturing a 3D porous medium that has the potential to utilize motile bacteria to move toward and break down contaminants that are trapped in soils, sediments, and subsurface formations (Datta & Bhattacharjee, 2020). Tianjin University has filed a patent for a high-affinity and high-mechanical double-network printing ink that enables the creation of a high-functional 3D microbial material with improved bioremediation efficiency and resistance to complex environmental impacts (Zhao et al., 2022). These case studies, complemented by patents from leading universities, illustrate the burgeoning role of 3D printing in bioremediation. They not only validate the efficacy and commercial potential of these technologies but also highlight the synergy between engineering and microbial ecology. This conjunction is paving the way for innovative, effective, and adaptable environmental remediation strategies, marking a significant advancement in the application of 3D bioprinting technologies for ecological restoration.

Challenges and limitations

Despite promising advancements, integrating 3D printing in bioremediation faces several challenges. One of the main concerns is the economic feasibility, particularly the high costs associated with certain 3D printing technologies and the research-intensive material development phase, which might be time and cost-prohibitive and may not be justified by the scale of many bioremediation projects (Ngo et al., 2018). Scalability remains a hurdle, as translating laboratory-scale successes to field applications is often challenging due to the complexities of real-world environmental conditions (Park et al., 2022). Sustainability issues also arise, especially in the life cycle assessment of the materials used for printing, focusing on the energy consumption and potential waste generated by the printing process (Nadagouda, Ginn & Rastogi, 2020). Despite the challenges, researchers continue to work towards overcoming these obstacles and advancing the use of 3D printing in bioremediation, and the future looks promising.

Future directions

In this section, we explore the possibilities of using new materials and methods that incorporate advanced technologies to tackle evolving challenges and opportunities. We investigate the potential synergy between 3D bioprinting and emerging fields and how they can possibly be leveraged to create innovative solutions.

Advanced materials

Exploring advanced materials in 3D printing holds significant promise for enhancing bioremediation strategies. Smart polymers that respond to environmental stimuli such as pH, temperature, or the presence of specific contaminants could revolutionize the way bioremediation is approached by enabling more dynamic and responsive cleanup processes. Nanomaterials, such as nanoparticles with catalytic properties, can also be integrated into 3D-printed structures to boost the efficiency of pollutant degradation; however, the impact of using such particles needs to be assessed from a sustainability perspective. Research into biodegradable and bio-based printing materials further aligns with the sustainability goals of bioremediation, minimizing the environmental footprint of the remediation tools themselves (Wei et al., 2017; Shafranek et al., 2019). Researchers are also investigating the 3D printability of algae-based materials and have found that PHAs derived from algae could be a sustainable alternative while maintaining excellent mechanical properties and being environmentally friendly (Grira et al., 2023). Researchers are exploring the use of 3D printing to create eco-friendly geopolymer materials that can remove methylene blue from wastewater. These materials made using reduced graphene oxide (rGO) and zinc oxide (ZnO) had achieved an impressive 92.56% removal efficiency of MB within just 30 min, and they used the same geopolymer ink, which contained 56% rGO@ZnO, to 3D print a scaffold using Direct Ink Writing technology (Liu et al., 2023a). These newer materials can further help improve the field by improving the multifunctionality of the constructs while enhancing their robustness, durability, and environmental sustainability. Utilizing these materials can potentially increase the efficiency and effectiveness of bioremediation processes, ultimately leading to a cleaner and healthier environment.

Integration with other technologies

Integrating 3D printing with the Internet of Things (IoT) and artificial intelligence (AI) presents exciting opportunities to create more intelligent and autonomous bioremediation systems. IoT devices can provide real-time monitoring of environmental conditions and pollutant levels, feeding data into AI algorithms to predict and adjust the bioremediation process for improved results. The potential for self-regulating bioremediation systems, which adapt to changing conditions without human intervention, could be realized through the convergence of these technologies, significantly increasing the efficacy and reducing the cost of bioremediation operations (Lawless et al., 2019; Salam, 2020). Machine learning is also being explored to optimize processes, applied materials, and biomechanical performances to enhance bioprinting and bioprinted constructs (Sun et al., 2023) and someday could help tailor 3D printable materials specific to the contaminated sites. Bioprinting in space missions to produce engineering living materials capable of oxygen production and wastewater treatment could significantly impact the development of bioregenerative life support systems (Krujatz et al., 2022). Additive manufacturing has the potential to contribute significantly to the field of bioremediation for terraforming applications and one day might play a pivotal role in making the atmosphere, volatile components, temperature, surface topography, or ecology of astronomical bodies habitable for human settlement. Automation of bioprinting processes coupled with robotic platforms brings a new dimension of functionality to the field of bioremediation. As suggested by one study, incorporating an advanced artificial intelligence-based control system into in situ bioremediation of petroleum-contaminated groundwater systems significantly improved the efficiency and effectiveness of a process, leading to better remediation results (Hu, Huang & Chan, 2003). As 3D printing technology continues to evolve, it could be used to produce customized bioremediation systems that incorporate advanced AI-based control mechanisms, leading to more effective and efficient remediation outcomes.

Policy and regulation

Supportive policy and regulatory frameworks are essential for 3D printing technologies to become a mainstay in environmental management. Policies encouraging research and development and incentives for adopting green technologies can accelerate the integration of 3D printing in bioremediation. Regulations will need to evolve to ensure the safe deployment of these technologies, especially considering the use of novel materials and the potential generation of byproducts from 3D printing processes (Papaconstantinou & Polt, 1997; Baiano, 2022). The European Union, for example, funds multiple water security projects that lead to the widespread implementation of novel solutions and innovation (Community Research and Development Information Service (CORDIS), 2020).

Conclusion

3D printing technology stands on the brink of revolutionizing bioremediation, offering unparalleled precision in fabricating structures that support intricate microbial ecosystems, enhance enzyme stability, and facilitate heavy metal sequestration. Nevertheless, while its potential is profound, several critical challenges and questions remain unaddressed, casting a shadow on the path to its widespread implementation. One of the most pressing issues lies in the economic viability of upscaling 3D-printed bioremediation solutions. Currently, the costs associated with 3D printing advanced materials, particularly at the scale required for impactful environmental applications, are not insignificant. This economic barrier must be surmounted to enable broader adoption of these technologies. Moreover, scalability extends beyond cost to the technical challenges of producing and deploying large-scale bioremediation structures in diverse environmental contexts.

Regulatory frameworks also lag behind technological advancements, with current policies often ill-equipped to manage the nuanced risks and benefits of deploying 3D-printed materials in ecological settings. The development of comprehensive regulations that both promote innovation and ensure environmental safety is a critical need that must be met to foster public trust and industry growth. Looking to the future, unanswered scientific questions beckon for research into the long-term stability and functionality of 3D-printed bioremediation systems. The environmental impact of these materials, the degradant byproducts that they might produce, and the potential for nanoparticle or chemical leaching present a significant gap in our current understanding. Furthermore, while integrating IoT and AI holds promise for real-time monitoring and responsive bioremediation strategies, the practicalities of such systems under variable environmental conditions are yet to be fully explored.

In conclusion, the pathway for 3D printing in bioremediation has immense potential and adoptive challenges. As this technology advances, it is imperative that research continues to address these economic, scalability, and regulatory challenges, as well as the pressing environmental safety and technical questions. Only through a concerted effort to bridge these gaps can we harness the full potential of 3D printing, steering the future of bioremediation toward more intelligent, effective, and sustainable practices. As the field of bioremediation continues to evolve, it is becoming increasingly clear that 3D printing has a crucial role to play in the development of more sophisticated and effective remediation technologies. By taking advantage of the unique capabilities of 3D printing, researchers and engineers can create highly customized and precise structures that can optimize the delivery of remediation agents to contaminated sites. Furthermore, 3D printing can be used to create complex microenvironments that mimic the natural conditions of soil and groundwater, allowing for more accurate testing and validation of new remediation techniques. As a result, 3D printing is poised to revolutionize the way we approach bioremediation, unlocking new opportunities for sustainable environmental management and protection.

Additional Information and Declarations

Competing Interests

Author Contributions

Data Availability

Abraham Samuel Finny is employed by Waters Corporation.

Abraham Samuel Finny conceived and designed the experiments, performed the experiments, analyzed the data, prepared figures and/or tables, authored or reviewed drafts of the article, and approved the final draft.

The following information was supplied regarding data availability:

This is a literature review.

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
