# Peer review of "D bioprinting in bioremediation: a comprehensive review of principles, applications, and future directions"

_PeerJ, doi:10.7717/peerj.16897_

## Round 0.1 · original submission · Major Revisions

· Academic Editor

Major Revisions

the work has the potential to be published. Please make the necessary corrections and send them to us.

Reviewer 1 ·

Basic reporting

I congratulate the author for his wide data set. In general, the manuscript is written in professional language.
The introduction adequately presents the subject and the motivation of the review.
The literature is well referenced and relevant.
The manuscript is within the scope of the journal and presents a broad and cross-disciplinary interest
There is a review of the same field from 2021, but Finny review present a different point of view and describe more actual bioprinting techniques.
English language should be improved, particularly in two sections: all over the abstract and in lines 42-50. This should be improved upon before acceptance.

Experimental design

The manuscript structure follows PeerJ standards. Nevertheless, title font size should be unified. For example, title in line 170 seems bigger…

Validity of the findings

The Conclusion identifies unresolved future directions

·

Basic reporting

The article seems to be scientifically sound and giving significant contrbution in the field of 3D printing techniques. Following comments are needed to be addressed.
1. The bibliometric analysis could be provided to visualise the trend of literatures and other aspects in the direction of 3D printing techniques and its applications.
2. The 3D printing techniques i.e., SLA, SLS, FDM, etc. are well known and reported earlier, these are needed to be concised.
3. The applications parts are needed to be more detailed as lots of applications in the field of advanced wastewater treatment are there, which are needed to be addressed.
4. The Review article are not only the reporting, this should be in analytical way and should have recommendations for future research.

Experimental design

Study design is needed to be improved as lots of repetetive informations are needed to be removed and more applications are to be added. Analysis of the results are needed to give a realization of the results.

Validity of the findings

The findings may be more informative and a pathways for young researchers, hence, analysis of the results are needed to be carried out.

Additional comments

1. Analysis of the results are needed.
2. More depth recommendations for the future research.
3. More Figures are needed to be added on applications and analysis parts.
4. Here analysis means the removal potential or other applications are needed to be analysed on the basis of some stats like removal efficiency, no. of applications, relations with cost and manufacturing time, etc.

Reviewer 3 ·

Basic reporting

the review is well structured and clearly written with comprehensive literature reviews. The topic of the review is timely and presents inspiring views to the community of additive manufacturing and environmental science

Experimental design

The author presents a comprehensive review of the integration of 3D printing/bioprinting technologies in bioremediation. This review highlights the key considerations in the fusion of engineering and material for the application of environmental restoration. It includes basic principles and definitions, applications, and future recommendations

Validity of the findings

The review adeptly connects its conclusions to the initial research questions, despite the inapplicability of experimental findings or impact assessments. It thoughtfully addresses future research directions

Additional comments

However, I do encourage the author to consider some necessary changes before it can be accepted for publication.

1. In the "What is 3D Bioprinting?" section (line 160), the current definition from the author is somewhat simplistic and facile. To enhance this section, I will consult the suggested papers to formulate a more comprehensive and widely recognized definition of 3D bioprinting. This could capture the full scope of 3D bioprinting so that the definition aligns with the consensus in the scientific community.

a. Murphy SV, Atala A. 3D bioprinting of tissues and organs. Nature biotechnology. 2014 Aug;32(8):773-85.
b. Hölzl K, Lin S, Tytgat L, Van Vlierberghe S, Gu L, Ovsianikov A. Bioink properties before, during and after 3D bioprinting. Biofabrication. 2016 Sep 23;8(3):032002.
c. Fu Z, Ouyang L, Xu R, Yang Y, Sun W. Responsive biomaterials for 3D bioprinting: A review. Materials Today. 2022 Jan 1;52:112-32.
d. Groll J, Burdick JA, Cho DW, Derby B, Gelinsky M, Heilshorn SC, Juengst T, Malda J, Mironov VA, Nakayama K, Ovsianikov A. A definition of bioinks and their distinction from biomaterial inks. Biofabrication. 2018 Nov 23;11(1):013001.

2. Section “Additive Manufacturing Techniques in Bioremediation” (line 268) presented a review of major 3D printing technologies used for bioremediation. Multi-material 3D Printing or 3D Bioprinting is a sub-section under this. This organization may lead to confusion, as multi-material printing is a broad technique applicable to both general 3D printing and bioprinting. To clarify this, I propose creating a distinct section solely dedicated to 3D bioprinting. This section would focus specifically on 3D printing involving microbials or cells that are encapsulated in hydrogels and biopolymers. Such an arrangement can align more closely with the keyword in the title.

3. I suggest the author integrate section “Material Selection and Sustainability” (Line 190) with “Integration with Microbial Systems” (Line 212). Compatibility of the materials with microbial systems shall be a criterion for material selection rather than a separate consideration.

4. For table 1, it would be better to add a column with the corresponding references of applications.

5. Figure 1, Figure 3, Figure 4, and Figure 6 each of them uses a different labelling format for subfigures. Please make sure you have consistent labelling for each of the figures.

6. The current formatting makes it challenging to distinguish between the title and subtitles due to their similar appearance. To enhance readability and clarity, I recommend adding numbering to the headings throughout the document. This modification will create an easily navigable layout. Additionally, please make sure such these changes are in alignment with the journal's formatting guidelines.

Annotated reviews are not available for download in order to protect the identity of reviewers who chose to remain anonymous.

---

## Round 0.2 · accepted · Accept

· Academic Editor

Accept

The author made all the suggested corrections, Congratulations.

Reviewer 1 ·

Basic reporting

The article is written in professional language.
The introduction adequately presents the subject and the motivation of the review.
The literature is well referenced and relevant.

Experimental design

The manuscript structure follows PeerJ standards.

Validity of the findings

The Conclusion identifies unresolved future directions

Additional comments

The author made all the suggested corrections

·

Basic reporting

The manuscript has been improved, as suggested.

Experimental design

Study design has been significantly improved.

Validity of the findings

Are in line and further improved.

Additional comments

Now the revised version could be accepted for publication.

Reviewer 3 ·

Basic reporting

NA

Experimental design

NA

Validity of the findings

NA

Additional comments

The author has addressed all my comments and the paper can now be accepted for publication